# Antifatigue Effects of 5-Aminolevulinic Acid Chronic Treatment on Mice

**DOI:** 10.3390/life15091465

**Published:** 2025-09-18

**Authors:** Chinatsu Ohmori, Eiko Kumamoto, Satoka Kasai, Kotaro Okano, Urara Ota, Atsuko Kamiya, Mitsugu Yamauchi, Kiwamu Takahashi, Masahiro Ishizuka, Kazumi Yoshizawa, Daisuke Yamada, Akiyoshi Saitoh

**Affiliations:** 1Laboratory of Pharmacology, Department of Pharmacy, Faculty of Pharmaceutical Sciences, Tokyo University of Science, 6-3-1 Niijyuku, Katsushika-ku, Tokyo 125-8585, Japanyamadada@rs.tus.ac.jp (D.Y.); 2Laboratory of Pharmacology and Therapeutics, Faculty of Pharmaceutical Sciences, Tokyo University of Science, 6-3-1 Niijyuku, Katsushika-ku, Tokyo 125-8585, Japan; satoka_kasai@rs.tus.ac.jp (S.K.);; 3SBI Pharmaceuticals Co., Ltd., Tokyo 106-6013, Japanatsuzuki@sbigroup.co.jp (A.K.);

**Keywords:** fatigue, monoamine, 5-aminolevulinic acid, protoporphyrin IX, MHPG, noradrenaline

## Abstract

5-aminolevulinic acid (5-ALA) is a heme precursor involved in mitochondrial activation. A clinical study suggested that 5-ALA supplementation alleviates fatigue in healthy individuals who experience chronic physical tiredness. However, the detailed mechanisms are unknown. Therefore, we investigated the mechanism underlying the antifatigue effect of 5-ALA using fatigue mouse models. C57BL/6N mice were orally administered 5-ALA hydrochloride or distilled water for 8 weeks. Fatigue mouse models were developed by housing the mice in a cage filled with water for 4 days. Fatigue was evaluated through running distance via a treadmill test. The decrease in the running distance in female mice significantly recovered after 5-ALA administration. 5-ALA administration ameliorated the decreased blood glucose levels in fatigue mouse models. These results suggest that 5-ALA improves fatigue-induced hypoglycemia by promoting the use of fatty acids. PpIX’s concentration in the FCX of the fatigue mouse models significantly increased after 5-ALA treatment. Decreased levels of 3-methoxy-4-hydroxyphenylglycol and noradrenaline (NA) turnover ratio in the FCX recovered to non-fatigue levels after 5-ALA treatment. Therefore, the antifatigue effect of 5-ALA in mice could be related to the activation of the NA neuronal systems in the FCX and the increase in energy production via glycogenesis activation from peripheral adipose tissue.

## 1. Introduction

5-aminolevulinic acid (5-ALA) is a heme precursor associated with mitochondrial activation. Heme is the central substance in the respiratory chain complex that produces energy during mitochondrial activation, which is the basis of life processes. 5-ALA is formed from glycine and succinyl coenzyme A (CoA) by 5-ALA synthase in mitochondria. The final enzyme complex IV or cytochrome c oxidase of the mitochondrial electron transfer chain is activated in the liver of mice chronically supplemented with 5-ALA [1].

Mitochondrial dysfunction is associated with chronic fatigue. A clinical study suggested that 8 weeks of 5-ALA phosphate supplementation alleviates fatigue in healthy individuals who experience chronic physical tiredness [2]. 5-ALA improves overall fatigue, reduces work-related fatigue, increases work efficiency, and lessens fatigue when waking up [2]. Interestingly, 5-ALA has beneficial effects in reducing negative moods associated with anger and hostility, suggesting its effects on the human central nervous system [2]. Furthermore, 5-ALA increases exercise efficiency and voluntary achievement during home-based walking training in healthy older women [3] and improves depression symptoms in middle-aged depressive women [4]. These studies reported that the increase in plasma lactate concentration, oxygen consumption rate, and carbon dioxide production rate during exercise was significantly attenuated in the 5-ALA treatment group, suggesting that 5-ALA supplementation improves mitochondrial function to recover the age-associated decrease in transient oxygen usage rates for aerobic adenosine triphosphate (ATP) production [5]. Thus, 5-ALA improves physical tiredness and the negative mood associated with fatigue in humans. However, the mechanisms underlying the antifatigue effects of 5-ALA remain unknown.

The monoaminergic system in the cortex is a principal effector of stress response. Prolonged exercise stress increases monoamine levels in the rat cortex [6,7]. In contrast to peripheral muscle fatigue, there is a concept of fatigue known as central fatigue, which arises from impaired function of the central nervous system, including the brain and spinal cord. Matsui et al. (2011) [8] reported that exercise increases the activation of monoamine metabolism, including norepinephrine, in the cortex and causes a decrease in brain glycogen. This suggests that changes in monoamine metabolism serve as an indicator of central fatigue [8]. Furthermore, NA and its metabolite 3-methoxy-4-hydroxyphenethylene glycol (MHPG) in the cerebral cortex and locus coeruleus regions are altered by excessive stress [9,10]. Another study reported a significant correlation between MHPG levels and somatization disorder in patients with major depression [11]. Furthermore, monoamine oxidase (MAO) inhibitors may be effective for alleviating central fatigue in bipolar depression [12]. Thus, changes in the activation of the noradrenergic system in the prefrontal cortex reflect central fatigue, stress, and depression.

Therefore, we hypothesized that the antifatigue effects of 5-ALA are associated with increased energy production in the periphery or activation of the NA neuronal system in the frontal cortex (FCX). Therefore, to clarify the mechanism underlying the antifatigue effects of 5-ALA, animal models of chronic fatigue were established [13], and the effects of 5-ALA on the NA monoaminergic function (NA metabolic activity and MAO activity) of the FCX and fat weight were examined.

## 2. Materials and Methods

### 2.1. Animals

Female and male C57BL/6N mice (6 weeks old) were purchased from CLEA Japan (Tokyo, Japan). A total of 191 mice were used in this study. The animals were housed in groups of six per cage at 23 °C ± 1 °C, and food was provided ad libitum. Furthermore, 12 h light–dark cycle was maintained (lights on 8:00–20:00 h). The experimental protocols were approved by the Institutional Animal Care and Use Committee of Tokyo University of Science (approval number: Y24006), and this study was conducted according to the guidelines of the National Institute of Health and the Japan Neuroscience Society.

### 2.2. Drugs

The mice were orally administered 5-ALA hydrochloride (SBI pharmaceuticals Co., Ltd., Tokyo, Japan) dissolved in drinking distilled water (DW). Then, they were given 5-ALA hydrochloride at a dose of 300 mg/kg body weight/day for 8 weeks (4.62 ± 0.52 mL/day drinking water containing 1.2 mg/mL 5-ALA). DW was added to drinking water of the control group. The dosage for mice in this study was determined by converting the human equivalent dose (HED), as outlined in our earlier publications [14]. In pharmaceuticals, the HED is commonly used to calculate the appropriate dose for test animals, considering their body surface area, to achieve effects similar to those in humans. Consequently, the dose of 300 mg/kg/day administered to mice equates to a human equivalent dose (HED) of 24 mg/kg/day, which is within the safety parameters specified in the 5-ALA interview form, which provided by the pharmaceutical company in Japan. Mice were allowed free access to Labo MR Stock (NOSAN LABO Series) feed. This feed contains iron at 239.8 mg/kg.

### 2.3. The Animal Model of Fatigue

All the groups used for this experiment were DW-treated fatigued/no-fatigued and 5-ALA treated fatigued/no-fatigued mice. The animal model of fatigue was developed by filling cages with water (23 °C ± 1 °C) to a height of 0.5 cm (water cage). We used this previously described animal fatigue model with modifications [13]. Fatigue-like behavior of mice treated with 5-ALA for 8 weeks was then assessed using treadmill exercise as an index of whole-body exercise capacity, as previously mentioned [15]. After 8 weeks of the 5-ALA treatment, the mice were subjected to treadmill pretest. After the treadmill pretest, the mice were kept in water cages for 4 days and were used as animal fatigue models. During the treadmill post-test (Day 5), the mice were resubjected to the same treadmill protocol (Table 1). The workload was calculated as the percentage change product of the running distances until exhaustion between the pretest (Day 1) and post-test (Day 5). The 5-ALA treatment intervention was performed as a pretreatment to induce fatigue. The intervention was also continued to be given while inducing fatigue.

The speed was gradually increased by 2 m/min every 5 min until the exhaustion. The criterion for exhaustion was that the mice unable keep running and rest in the posterior 1/2 of the runway for 3 s.

### 2.4. Treadmill Fatigue Test

The treadmill fatigue test was performed according to a previous report [16]. During the treadmill test, each mouse was forced to run on a motor-driven treadmill (TMS-2; Melquest, Toyama, Japan). At the treadmill pretest (Day 1), the mice were subjected to the treadmill protocol (Table 1). The animals that ran > 460 m (26 m/min × 5 min) and <1000 m (34 m/min × 5) during the pretest were selected for analysis in this study. First, a 10 min warmup period was provided with the treadmill set at 10–15 m/min and 0° inclination. After warmup, the treadmill inclination was fixed at 10°. The test was started at a speed of 20 m/min, and the speed was increased by 2 m/min every 5 min until the mouse reached exhaustion. Exhaustion was defined as spending 3 s on the shocker plate without attempting to re-engage the treadmill.

### 2.5. Measurement of Adipose Tissue Mass and Blood Glucose Levels

Immediately after the treadmill fatigue test, the mice were decapitated, and both the visceral adipose tissue around the ovary and blood samples was collected. General anesthesia was not used on the mice. Wet weights of adipose tissues were then measured. The blood glucose levels were measured using a One Touch Verio Reflect blood glucose monitoring system (LifeScan Japan Co., Ltd., Tokyo, Japan).

### 2.6. Measurement of Monoamine Levels

Immediately after the treadmill fatigue test, the mouse brain was dissected, and the FCX was fractionated. The brain tissues were stored at −80 °C until use. MHPG and NA levels were quantified using high-performance liquid chromatography (HPLC). The brain tissues were homogenized in 300 μL of 0.2 M perchloric acid containing 100 ng of isoproterenol as an internal standard. The homogenates were placed on ice for 30 min and then centrifuged at 15,000 rpm for 20 min at 4 °C. To maintain the mixture at pH 3, 0.1 M sodium acetate was added to the supernatant. The sample was filtered using 0.2 μm Minisart RC 4.0 (Sartorius, Tokyo, Japan). Ten-microliter samples were analyzed by HPLC via electrochemical detection. The electrochemical detector (ECD-300; Eicom Co., Ltd., Kyoto, Japan) was equipped with a graphite electrode (WE-3G; Eicom Co., Ltd., Kyoto, Japan) that was used at a voltage setting of 750 mV with an Ag/AgCl reference electrode [17]. The mobile phase comprised a 0.1 M sodium acetate/0.1 M citric acid buffer (pH 3.5) containing 12% methanol, 180 mg/L 1-octanesulfonic acid sodium salt, and 5 mg/L EDTA-2 Na. Monoamines were separated on a C-18 column (150 mm × 3 mm reversed-phase, EICOMPAK SC-5ODS, Eicom Co., Ltd., Kyoto, Japan). The mobile phase flow rate was maintained at 0.3 mL/min with a column temperature of 30.0 °C [17].

### 2.7. Measurement of MAO Activities in Mouse FCX

After 8 weeks of treatment with 5-ALA (1.2 mg/mL), the mouse brain was dissected, and the FCX was fractionated. The brain tissues were stored at −80 °C until use. Mitochondria were isolated using a mitochondria isolation kit for tissue (Thermo-Fisher Scientific, Waltham, MA, USA), according to the manufacturer’s instructions. MAO-A activities were evaluated using the MAO-Glo TM Assay Kit (Promega, Madison, WI, USA), according to the manufacturer’s protocol with a slight modification. Fractionated mitochondria were added as a test compound, and MAO enzyme solution was used as the standard. Protein concentrations were quantified using Bradford’s method (BCA TM Protein Assay Kit, Thermo-Fisher Scientific, Waltham, MA, USA).

### 2.8. Measurement of Protoporphyrin IX (PpIX) Levels

The concentration of 5-ALA in the FCX was quantified using the concentration of its metabolite PpIX as an indicator to confirm its translocation to the brain. Since 5-ALA is rapidly metabolized to PpIX, 5-ALA is hardly detected in the brain. After oral administration of 5-ALA, the mouse brain was dissected, and the FCX was fractionated. The brain tissues were stored at −80 °C until use. The PpIX concentration in the FCX of mice was determined using a previously described method [18], with appropriate modifications. In brief, the brain tissues were homogenized in 350 μL of phosphate-buffered saline at approximately eight times the tissue weight. 0.1 mL of homogenates were vigorously agitated for 30 s with 0.01 mL of 50% *v*/*v* acetic acid and 0.3 mL of N, N-dimethylformamide-2-propanol (DMF-IPA) solution (100:1 by vol.) and centrifuged at 17,700× *g* for 5 min at 4 °C to collect the supernatant. The precipitate was extracted with 0.15 mL of DMF-IPA solution by repeating the same procedures as above. The two batches of supernatant were mixed and analyzed using an HPLC system using a Capcell Pak C18 UG120 column (5 µm, 4.6 × 150 mm, Osaka Soda Co., Ltd., Osaka, Japan), mobile phase of acetonitrile-10 mM tetrabutylammonium hydroxide solution (pH7.5) (70: 30 by vol., flow rate, 1.0 mL/min; elution temperature, 40 °C), and fluorescence detector (Ex. 400 nm, Em. 630 nm). 

### 2.9. Statistical Analysis

The results are expressed as means ± standard error of the mean (SEM). Differences were evaluated using Student’s *t*-test or two-way analysis of variance (ANOVA), followed by the Holm–Sidak multiple comparisons test. All statistical analyses were performed using Prism 7 (GraphPad Software, San Diego, CA, USA). *p*-values < 0.05 were used to denote statistical significance. The *p*-values are shown as two-tailed values after correction for multiple comparisons.

## 3. Results

### 3.1. Mileage Ratio Assessment in the Treadmill Fatigue Test

Although no significant difference in the mileage ratio was observed between male and female mice in the no-fatigue mouse group (t (11) = 0.424; *p* = 0.679) (Figure 1A), the mileage ratio of female fatigued mice was significantly reduced compared with that in male fatigued mice (t (15) = −2.564; *p* = 0.021) (Figure 1B). Therefore, it was determined that all the following experiments should involve female mice only.

DW-treated female fatigued mice exhibited a significant reduction in the mileage ratio compared with DW-treated female mice without fatigue (Interaction, F(1,37) = 1.499; Drug, F(1,37) = 12.09; Fatigue, F(1,37) = 68.38) (Figure 2). In contrast, female fatigued mice administered with 5-ALA for 8 weeks showed a significant increase in the mileage ratio compared with DW-treated female fatigued mice (Figure 2). DW and 5-ALA administration had no effects on the mileage ratio in female mice without fatigue (Figure 2).

### 3.2. Body Weight, Adipose Mass, and Blood Glucose Levels

Two-way ANOVA indicated that the fatigued mouse group showed a significant decrease in body weight compared with the no-fatigue mouse group, whereas administration of 5-ALA had no effects on body weight (Interaction, F(1,37) = 0.07001; Drug, F(1,37) = 1.218; Fatigue, F(1,37) = 57.26) (Figure 3A).

Chronic drinking of 5-ALA significantly increased the fat weight in mice without fatigue compared with that in DW-treated mice without fatigue, whereas DW and 5-ALA treatment did not have effects mice with fatigue (Interaction, F(1,35) = 10.9; Drug, F(1,35) = 62.29; Fatigue, F(1,35) = 5.052) (Figure 3B).

DW-treated fatigued mice showed a significant decrease in blood glucose levels compared with DW-treated mice without fatigue (Interaction, F(1,40) = 4.821; Fatigue, F(1,40) = 79.65; Drug, F(1,40) = 2.813) (Figure 3C). However, 5-ALA administration significantly increased blood glucose levels in fatigued mice compared with that in DW-treated fatigued mice (Figure 3C).

### 3.3. Levels of Monoamine and Its Metabolite in the FCX of Fatigued Mice

No significant effects on NA level were observed in all groups (Figure 4A). The MHPG level (Figure 4B) and NA turnover ratio (Figure 4C) were significantly decreased in the DW-treated fatigued group compared with those in DW-treated mice without fatigue (Interaction, F(1,22) = 14.06; Drug, F(1,22) = 8.748; Fatigue, F(1,22) = 17.16) (Figure 4B) (Interaction, F(1,22) = 15.11; Drug, F(1,22) = 10.01; Fatigue, F(1,22) = 11.3) (Figure 4C). No effects on the MHPG level and NA ratio were observed in the DW-treated no-fatigued and 5-ALA-treated fatigued groups.

### 3.4. Measurement of MAO Activities in the FCX

MAO activity in 5-ALA-treated fatigued mice showed no statistical significance compared to saline-treated fatigued mice (*p* = 0.2997) (Figure 5). No changes were observed between 5-ALA- and DW-treated no-fatigued mice (Interaction, F(1,34) = 1.189; Drug, F(1,34) = 1.155; Fatigue, F(1,34) = 2.512) (Figure 5).

### 3.5. PpIX Levels in the FCX

The levels of PpIX, a metabolite of 5-ALA, in the FCX in 5-ALA-treated mice without fatigue was significantly increased compared with those in DW-treated mice without fatigue (Interaction, F(1,22) = 5.931; Drug, F(1,22) = 21.51; Fatigue, F(1,40) = 6.961) (Figure 6). However, no changes were observed between 5-ALA- and DW-treated fatigued mice.

## 4. Discussion

### 4.1. 5-ALA Has Antifatigue Effects on Fatigued Mice

In this study, the mileage ratio was significantly decreased because the mice were kept in breeding cages filled with water for 4 days (Figure 1A,B). These results were similar to those of a previous study [13], suggesting that these mice exhibit fatigue-like behavior. Although male mice were used as fatigue models in a previous study, in this study, the female mice exhibited greater fatigue than male mice among animal models of fatigue induced by breeding in water-filled cages (Figure 1B). These results are consistent with the finding that the incidence of chronic fatigue syndrome is threefold higher in women than in men [19]. Therefore, all subsequent experiments were performed using female mice.

The decrease in the mileage ratio improved after 5-ALA administration for 8 weeks (Figure 2), suggesting that 5-ALA has antifatigue effects. The mice were treated 8 weeks before fatigue induction; therefore, 5-ALA can prevent fatigue and alleviate fatigue. Clinical studies have suggested that 5-ALA supplementation can reduce fatigue in individuals with chronic physical tiredness [2], increase exercise efficiency [3], and alleviate depressive symptoms [4]. Our results are consistent with those of previous clinical studies on the antifatigue effects of 5-ALA. Previous clinical trials demonstrated that chronic treatment with 5-ALA produced, the anti-fatigue effects in healthy subjects who feel chronic physical tiredness suggesting 5-ALA has a potential as a therapeutic agent [2]. On the other hand, this study was conducted using a preventive protocol. We believe that future studies employing a therapeutic protocol are also warranted. 5-ALA improved exercise efficiency and home-based walking training achievement in older women [3] and improved depressive symptoms, as measured using the Montgomery–Åsberg Depression Rating Scale, in middle-aged depressive women [4]. Another clinical study suggested that 6-week 5-ALA administration enhanced sleep quality in individuals with insomnia or difficulty sleeping and reduced Pittsburgh Insomnia Rating Scale-20 Question scores [20]. These findings are similar to our findings that 5-ALA improves physical tiredness. The treadmill running test measures how long mice voluntarily maintain running. The mice are motivated to run until they cannot continue running on the treadmill to avoid further electric shocks. Thus, it measures mental fatigue (tiredness) and physical fatigue [15]. We suggested that the increased rate of change in the running test in the fatigued group suggests that 5-ALA improves the negative mood associated with fatigue (e.g., tiredness). The model used in this study also induces sleep deprivation. We consider it necessary to examine in future studies whether 5-ALA exerts anti-fatigue effects in other fatigue models, such as those created by forced swimming [21] or forced running tests [22].

### 4.2. 5-ALA Improves Fatigue-Induced Hypoglycemia by Promoting the Use of Fatty Acids

The adipose tissue weight in 5-ALA-treated mice without fatigue was significantly elevated compared with that in DW-treated mice without fatigue, whereas no significant effects on the adipose tissue weight were observed between 5-ALA- and DW-treated fatigued mice (Figure 3B). There are reports indicating that 5-ALA increases SREBP-1c expression, which in turn promotes the biosynthesis of fatty acids and triglycerides [23]. On the other hand, there are also reports suggesting that 5-ALA intake reduces fat [24]. Therefore, it remains unclear whether 5-ALA exerts a fat-increasing or fat-reducing effect. Regarding the present model, further investigation is warranted, including measurements of SREBP-1c, fatty acids, and triglycerides, to clarify these effects. Conversely, blood glucose levels significantly decreased in DW-treated fatigued mice. These decreased blood glucose levels improved in 5-ALA-treated fatigued mice; however, no changes in blood glucose levels were observed in 5-ALA- and DW-treated mice without fatigue (Figure 3C). Oral 6-week 5-ALA administration in mice with diet-induced obesity has been reported to reduce white adipose tissue weight and enhance the expression of mitochondrial oxidative phosphorylation complexes III, IV, and V in white adipose tissue [25]. This study suggested that the decrease in white adipose tissue caused by 5-ALA improves mitochondrial function [25]. White adipose tissue, a major energy storage tissue, is responsible for the synthesis and storage of triacylglycerol (TAG) during energy excess and for hydrolyzing TAG to produce fatty acids and glycerol for use by other organs during energy deficit [26]. Fatty acids are then metabolized via β-oxidation as acetyl-CoA to produce ATP in the mitochondrial electron transfer system (Figure 7). Glycerol is then converted to glucose via glycogenesis in the liver, which is metabolized in the glycolytic system to produce ATP (Figure 7). Taken together, we propose that peripheral 5-ALA promotes the use of fatty acids by activating the mitochondrial electron transfer system in adipose tissue, further producing fatty acids and glycerol in fatigued mice. Therefore, 5-ALA improves the hypoglycemic conditions of fatigued mice. Based on these findings, we propose that the antifatigue effects of 5-ALA are associated with enhanced metabolism in peripheral adipose tissue, which may contribute to the observed elevation in blood glucose levels.

### 4.3. 5-ALA Improves Fatigue-Induced Dysregulation of NA Systems in the FCX

Dysregulation of the central NA system has been proposed as a neural mechanism underlying some stress-related disorders, such as depression [27,28]. Quantification of monoamine levels revealed that MHPG levels in the FCX were significantly lower in the DW-treated fatigued group than in the no-fatigued group (Figure 4B). Therefore, the NA turnover ratio also decreased in the fatigued group. Conversely, oral 5-ALA administration reversed these changes, suggesting that 5-ALA restores the diminished MHPG levels during fatigue. MHPG is produced by the degradation of NA via MAO. Therefore, we hypothesized that the decrease in MHPG levels observed in the FCX of fatigued mice was caused by mitochondrial MAO dysfunction. Interestingly, although no effects on MAO activity in the FCX were observed in DW- and 5-ALA-treated mice in the no-fatigued group, 5-ALA-treated mice in the fatigued group exhibited a modest increase in MAO activity in the FCX compared with DW-treated mice. However, this difference did not reach statistical significance. Mitochondrial mass is increased in 5-ALA-treated mice [29], suggesting that increased mitochondrial mass in 5-ALA-treated mice in the fatigued group increased MAO activity. Our results suggest that the increase in mitochondrial mass in the FCX induced by 5-ALA could promote MAO activity, improving the effects on fatigue-induced NA dysregulation.

Another hypothesis is that the decrease in MHPG levels observed in fatigued mice may be due to decreased NA release from the presynapse of NAergic nerves. The activity of tyrosine hydroxylase (TH), which catalyzes the rate-limiting step in NA synthesis, changed the locus coeruleus and certain limbic regions, such as the hippocampus and amygdala, in rats exposed to short- and long-term forced walking stress [9,30]. Interestingly, rats exposed to short-term walking stress exhibited increased TH activity, whereas those exposed to long-term walking stress exhibited decreased TH activity and inactive depressed-like behaviors. Wang et al. suggested that the decrease in TH activity following long-term stress is caused by maladaptation to the stress response. Furthermore, Matsui et al. (2011) reported that MHPG levels in the rat cortex increased immediately after treadmill running for 120 min and were negatively correlated with glycogen, which is the principal energy source stored in the brain [8]. They suggested that changes in cortical MHPG levels can help understand central fatigue [8]. Moreover, individuals with depression and sleep disturbance were found to have significantly elevated MHPG levels in the cerebrospinal fluid [12], suggesting that excessive release of NA occurs during stress. Therefore, we proposed that chronic fatigue may have reduced NA release from the presynapse, decreasing MHPG levels. Conversely, oral 5-ALA administration may have maintained MHPG levels within normal ranges by suppressing the decrease in NA release during fatigue. Although the mechanism underlying the antifatigue effects of 5-ALA on the brain remains unclear, we propose that the improvement in fatigue-induced NA dysregulation in the FCX may be associated with a reduction in TH activity observed during fatigue. In the present study, histopathological evaluation of the prefrontal cortex, such as assessment of neuronal degeneration or loss, was not performed. Therefore, future studies addressing these aspects will be necessary.

### 4.4. 5-ALA Improves Fatigue-Induced Decreases in PpIX Concentration in the FCX

In this study, PpIX concentrations in the FCX significantly increased in control mice after 5-ALA treatment for 8 weeks compared with those in DW-treated mice (Figure 6). PpIX is produced by the binding of eight 5-ALA molecules. Our results suggest that 5-ALA could be absorbed in the brain and used for PpIX synthesis in the FCX of mice.

Interestingly, PpIX levels in the FCX of 5-ALA-treated fatigued mice reduced to the same levels as those in DW-treated fatigued mice after 5-ALA treatment discontinuation. Exogenously administered 5-ALA is metabolized to heme via the assembly of eight 5-ALA molecules into PpIX, a heme precursor, which is then inserted with a ferrous ion. Heme is the active center of the mitochondrial respiratory chain complex. Thus, heme deficiency and dysregulation of heme metabolism are important in neurodegenerative processes, suggesting that heme recovery is a preventable component [31]. Therefore, we propose that PpIX, increased by 5-ALA administration, reverses NA nerve dysfunction in the mouse FCX observed during fatigue. Therefore, we propose that repeated oral 5-ALA administration can restore mitochondrial function by increasing heme synthesis in the brain. We consider it necessary to measure changes in complex activity and brain ATP production following oral administration of 5-ALA in future studies.

Whether the peripheral and central effects of 5-ALA are correlated independently with each other is difficult to determine. 5-ALA alleviates chronic physical fatigue in healthy individuals [2]. Conversely, 5-ALA also increases motivation and voluntary achievement for exercise training in older women [3]. These results suggest that 5-ALA is effective for managing peripheral and mental fatigue. Further studies are necessary to clarify the ameliorative effects of 5-ALA on mental and physical fatigue.

## 5. Conclusions

We suggest that the antifatigue effects of 5-ALA on mice could be related to the improvement of fatigue by the activation of the NA neuronal systems of the prefrontal cortex and the increase in energy production by the activation of glycogenesis from peripheral adipose tissue. We propose that 5-ALA supplementation could improve not only physical tiredness but also negative mood associated with fatigue. This study was conducted using female mice, as chronic fatigue syndrome is more prevalent in women. However, sex differences observed in humans may not necessarily translate to mice, and future studies including male mice will be necessary to fully assess potential sex-related effects.

## Figures and Tables

**Figure 1 life-15-01465-f001:**
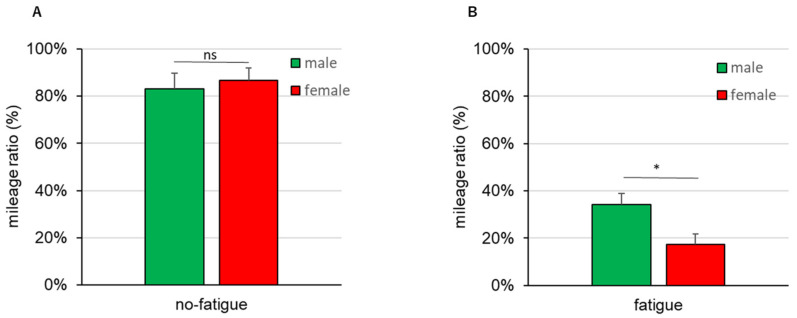
Effects of sex differences in the mileage ratio in the treadmill fatigued test. The treadmill fatigue test was performed according to our previous report [16]. At the treadmill pretest (Day 1), both fatigued and non-fatigued mice were subjected to the treadmill protocol (Table 1). After the treadmill pretest, non-fatigued mice were kept for 4 days in normal cages (**A**), while fatigued mice were kept for 4 days in cages filled with water (23 °C ± 1 °C) to a height of 0.5 cm (**B**). At the treadmill post-test (Day 5), fatigued and non-fatigued mice were resubjected to the same treadmill protocol. The workload was calculated as the % changes product of the running distance until exhaustion between pretest (Day 1) and post-test (Day 5). ns: not significant; * *p* < 0.05 vs. male mice. (Student’s *t*-test) Male/non-fatigue *n* = 6; Male/fatigue *n* = 9; Female/non-fatigue *n* = 10; and Female/fatigue *n* = 8.

**Figure 2 life-15-01465-f002:**
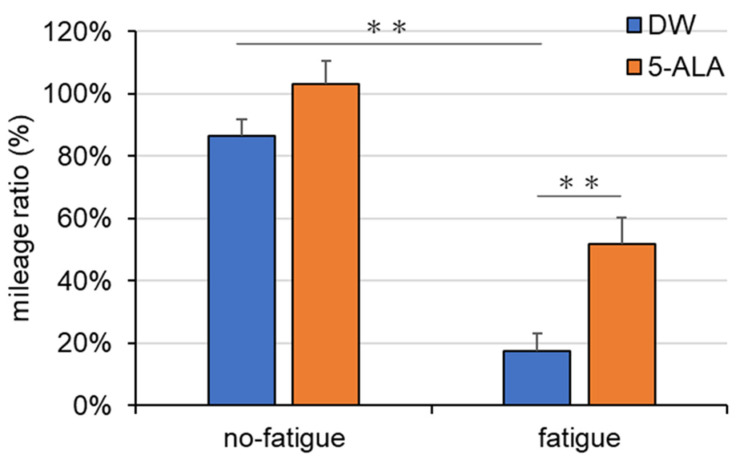
Effects of 5-ALA treatment on the mileage ratio in fatigued mice. After the administration of DW or 5-ALA, each group was kept in a normal cage (no-fatigued group) or in a cage filled with water (23 °C ± 1 °C) to a height of 0.5 cm (water cage) for 4 days (fatigue group). The treadmill fatigue test was performed on days 1 and 5. Data are represented as means ± SEM. (*n* = 6–12) ** *p* < 0.01 vs. DW-treated no-fatigued and fatigued group, respectively (two-way ANOVA followed by the Holm–Sidak multiple comparison test). DW/non-fatigue, *n* = 10; DW/fatigue, *n* = 8; 5-ALA/non-fatigue, *n* = 12; and 5-ALA/fatigue, *n* = 11.

**Figure 3 life-15-01465-f003:**
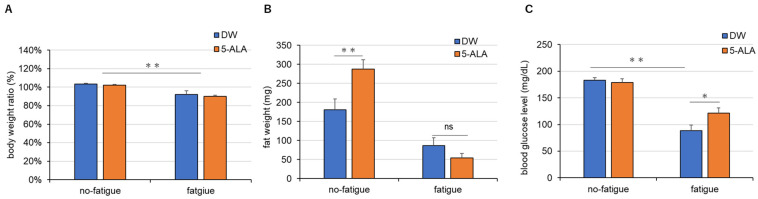
Effects of 5-ALA treatment on the weight (**A**), fat weight (**B**), and blood glucose levels (**C**) in fatigued mice. After the treadmill fatigue test on day 5, the weight of visceral adipose tissue around the ovary and blood glucose levels were measured. Data are represented as means ± SEM. (*n* = 8–12) * *p* < 0.05, ** *p* < 0.01 (two-way ANOVA followed by the Holm–Sidak multiple comparison test). ns: not significant (vs. DW-treated fatigue group). DW/non-fatigue, *n* = 6; DW/fatigue, *n* = 8; 5-ALA/non-fatigue, *n* = 6; and 5-ALA/fatigue, *n* = 6.

**Figure 4 life-15-01465-f004:**
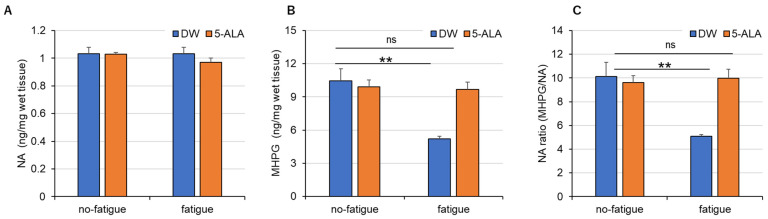
Effects of 5-ALA treatment on NA contents (**A**), its metabolite MHPG contents (**B**) and NA turnover ratio (**C**) of FCX in fatigued mice. Immediately after the treadmill fatigue test on day 5, the mouse frontal cortex (FCX) was quickly dissected. The levels of noradrenaline (NA) and 3-methoxy-4-hydroxyphenyl glycol (MHPG) were measured as a concentration per wet tissue weight using the HPLC-ECD system. Data are presented as means ± SEM. (*n* = 6–10). ns: not significant, ** *p* < 0.01 vs. DW-treated no-fatigued group (two-way ANOVA followed by the Holm–Sidak multiple comparison test). DW/non-fatigue, *n* = 10; DW/fatigue, *n* = 11; 5-ALA/non-fatigue, *n* = 10; and 5-ALA/fatigue, *n* = 7.

**Figure 5 life-15-01465-f005:**
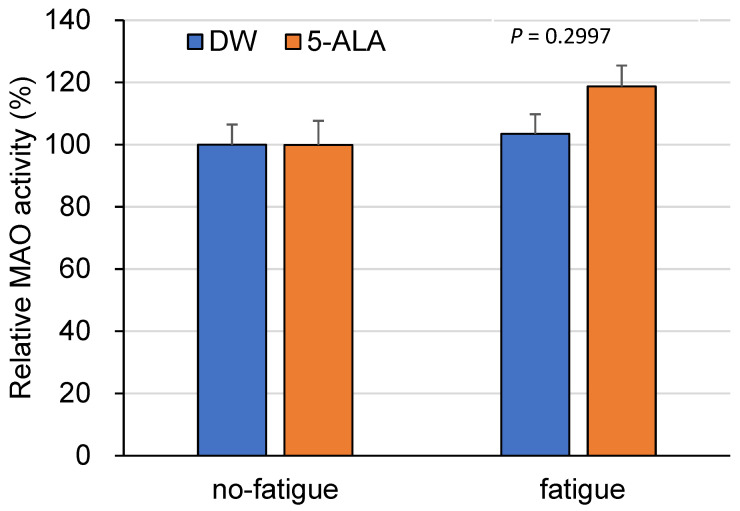
Effects of chronic 5-ALA treatment on MAO activity in the FCX of fatigued mice. After oral ad-ministration of 5-ALA (1.2 mg/mL) for 8 weeks, the mouse brain was dissected, and the frontal cortex (FCX) of mice was fractionated. The monoamine oxidase (MAO) activity was measured as a RLU per wet tissue weight. Data are presented as means ± SEM. (*n* = 7–11) (two-way ANOVA followed by the Holm–Sidak multiple comparisons test). DW/non-fatigue, *n* = 7; DW/fatigue, *n* = 8; 5-ALA/non-fatigue, *n* = 6; and 5-ALA/fatigue, *n* = 6.

**Figure 6 life-15-01465-f006:**
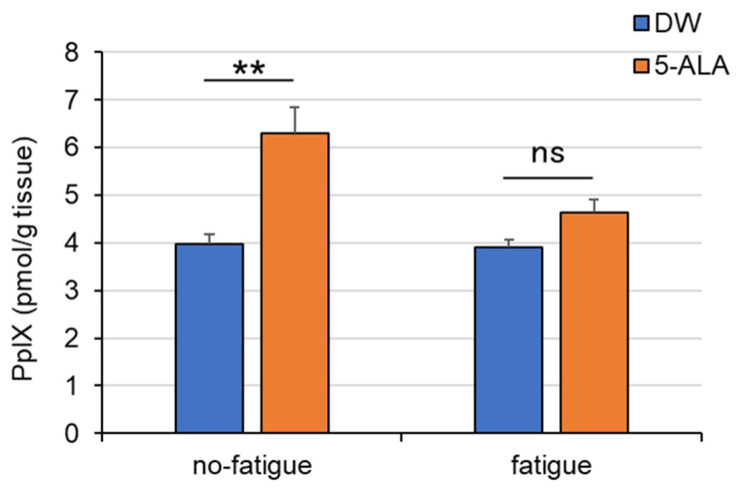
Effects of chronic 5-ALA treatment on PpIX in the FCX of fatigued mice. After oral administration of 5-ALA (1.2 mg/mL) for 8 weeks, the mouse brain was dissected, and the frontal cortex (FCX) of mice was fractionated. Data are presented as means ± SEM. (*n* = 6–8) ns: not significant. ** *p* < 0.01 (two-way ANOVA followed by the Holm–Sidak multiple comparisons test). DW/non-fatigue, *n* = 6; DW/fatigue, *n* = 6; 5-ALA/non-fatigue, *n* = 6; and 5-ALA/fatigue, *n* = 8.

**Figure 7 life-15-01465-f007:**
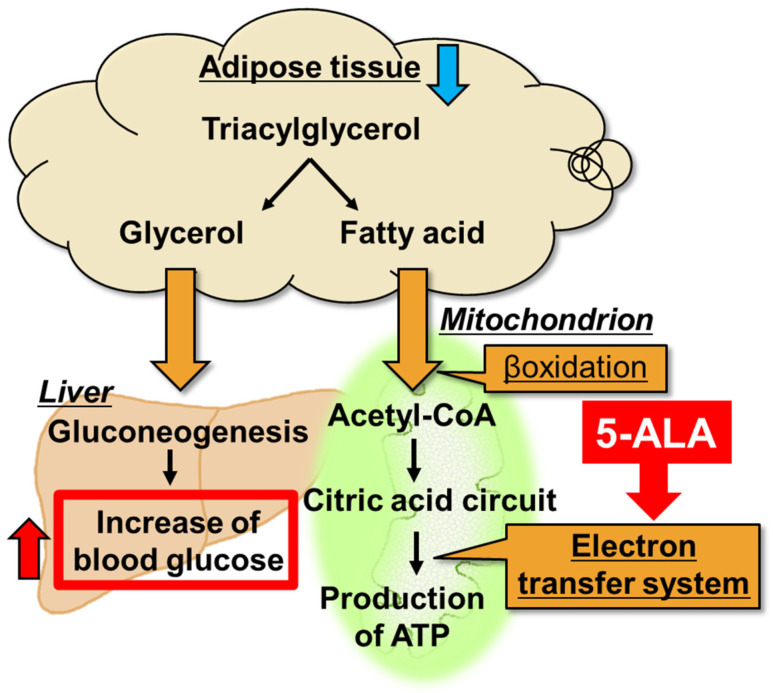
The mechanism of 5-ALA induced improvement of hypoglycemia in fatigued mice. 5-ALA decreases fat weight by activating the mitochondrial electron transfer system in adipose tissue and increases β-oxidation [25]. β-oxidation activation of fatty acid increases the differentiation of triacylglycerol to glycerol in white adipose tissue. The resulting increase in gluconeogenesis in the liver increased blood glucose levels and ameliorated hypoglycemia during fatigue.

**Table 1 life-15-01465-t001:** Treadmill fatigue test protocol.

	Speed (m/min) × Time (min)	Inclination
Warm-up	10 m/min × 5 min	0°
	15 m/min × 5 min	
Test	20 m/min × 5 min	10°
	22 m/min × 5 min	
	24 m/min × 5 min	
	26 m/min × 5 min	
	28 m/min × 5 min	
	30 m/min × 5 min	
	32 m/min × 5 min	
	34 m/min × 5 min	

## Data Availability

The original contributions presented in this study are included in the article. Further inquiries can be directed to the corresponding author.

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
