# Peer review of "Antifatigue Effects of 5-Aminolevulinic Acid Chronic Treatment on Mice"

_life, 2025, doi:10.3390/life15091465_

Round 1

Reviewer 1 Report

Comments and Suggestions for Authors

Dear authors. The article is devoted to a significant problem - the study of the possibility of correcting fatigue with 5-aminolevulinic acid. The article contains a sufficient amount of experimental material and is well executed. However, there are general observations: the administration needs to be processed, the design is not standardized, the number of animals in the groups is different (no justification is given for this), the 5-ALPHA administration regimen is not justified, there is no control of the administered dose, the data are presented in relative units, which distorts the results.The details of the remarks are in the attached file. As a strong point of the article, I would like to note the discussion of the results and logical conclusions.

Author Response

Comments1

The transition is abrupt and incomprehensible. Lead the reader to the central mechanisms of fatigue development.

It’s debatable. This assumption does not follow from the information presented above, which compromises the claimed theory.

Response1

Thank you for pointing that out. As suggested by reviewer 1, we added the comment concerning that the noradrenergic system in the prefrontal cortex reflects the central mechanisms of fatigue development, as follows (page2, Introduction Line 57, 68).

“In contrast to peripheral muscle fatigue, there is a concept of fatigue known as central fatigue, which arises from impaired function of the central nervous system, including the brain and spinal cord. Matsui et al. 2011) reported that exercise increases the activation of monoamine metabolism, including norepinephrine, in the cortex and causes a decrease in brain glycogen. This suggests that changes in monoamine metabolism serve as an indicator of central fatigue.”

“Thus, changes in the activation of the noradrenergic system in the prefrontal cortex reflect central fatigue, stress, and depression.”

Comments 2

Given the free use of 5-ALA by animals, the question arises how were the doses controlled and what concentration did

Response 2

Thank you for your suggestion. At the beginning of the experiment, we measured the reduction in water volume of each bottle in the cages to estimate the water intake per mouse. We have revised the description based on these measurements. Specifically, we changed the expression in the text from “approximately 5 mL/day” to “4.62 ± 0.52 mL/day” in the Materials and Methods section: (Page 2, 2.2 Line 91).

Comments 3

Please indicate whether a general anesthetic was used.

Response 3

Thank you for pointing this out. Therefore, we added the following sentences in the Materials and Methods section: (Page4,2.5 Line137)

“General anesthesia was not used on the mice.”

Comments 4

Back it up with a reference. The data is known.

Response 4

Thank you for your feedback. As you pointed out, the content overlapped with the section on Materials and Methods, so it has been removed. (page4, 2.8. Line175)

Comments 5

Provide absolute data. Replace it with running time. the data are presented in relative units, which distorts the results.

Response 5

Thank you for your valuable comment. In the present treadmill test, the ratio of the running distance measured before and after the water immersion-induced fatigue test was calculated in the same mice. Accordingly, the data are expressed in relative units.

Comments 6

This is typical for humans. Replace (line 297)

Response 6

Thank you for your comment. We acknowledge that sex differences observed in humans do not necessarily translate directly to mice. In this study, female mice were used because the disease of interest shows higher prevalence in women. We recognize that this choice does not fully model human sex differences, and we have revised the manuscript to clarify this limitation.

“This study was conducted using female mice, as chronic fatigue syndrome is more prevalent in women. However, sex differences observed in humans may not necessarily translate to mice, and future studies including male mice will be necessary to fully assess potential sex-related effects.” We added these comments in the Conclusions section (Page 12 line 444).

Reviewer 2 Report

Comments and Suggestions for Authors

Minor Revision

Authors present a study on 5-ALA's antifatigue effects in a mouse model. They used behavioral tests and biochemical assays to support your claims. The hypothesis links mitochondrial function and noradrenergic systems to fatigue relief.

1] Iron supplementation omitted, unlike human trials pairing 5-ALA with iron. This limits translation, as iron aids heme synthesis.

2] Study lacks direct mitochondrial function tests, like ATP production or complex activity. Rely less on indirect PpIX measures.

3] Fatigue model uses shallow water, inducing sleep deprivation over physical fatigue. Validate against other models, like forced swimming.

4] Define exhaustion inconsistently. Table says 3 seconds rest, text says 5 seconds. Standardized criteria.

5] The non-significant MAO changes are called "modest". Report them as non-significant.

6] Dose is too high for mice 300 mg/kg/day. Typical Human dose is 0.5-1.7 mg/kg. Test lower doses for translational value.

Author Response

Comments 1

Iron supplementation omitted, unlike human trials pairing 5-ALA with iron. This limits translation, as iron aids heme synthesis.

Response 1

Thank you for pointing this out. We added the following sentences regarding iron supplementation in the Materials and Methods section: (page 3, 2.2 Line 100)

“Mice were allowed free access to Labo MR Stock (NOSAN LABO Series) feed. This feed contains iron at 239.8 mg/kg.”

Comments 2

Study lacks direct mitochondrial function tests, like ATP production or complex activity. Rely less on indirect PpIX measures.

Response 2

Thank you for your comment. As pointed by reviewer, “We consider it necessary to measure changes in complex activity and brain ATP production following oral administration of 5-ALA in future studies.”. We added these comments in the Discussion section (Page 12 line 431).

Comments 3

Fatigue model uses shallow water, inducing sleep deprivation over physical fatigue. Validate against other models, like forced swimming.

Response 3

Thank you for your comment. As pointed by reviewer, “The model used in this study also induces sleep deprivation. We consider it necessary to examine in future studies whether 5-ALA exerts anti-fatigue effects in other fatigue models, such as those created by forced swimming [21] or forced running tests [22].” We added these comments in the Discussion section: (page 9, 4.1 Line 331)

Comments 4

Define exhaustion inconsistently. Table says 3 seconds rest, text says 5 seconds. Standardized criteria.

Response 4

Thank you for pointing that out. Three seconds is the precise requirement. We have corrected the wording in the Materials and Methods section. (page 4, 2. 4. Line 132)

Comments 5

The non-significant MAO changes are called "modest". Report them as non-significant.

Response 5

Thank you for pointing that out. We have revised the text in the Results section as follows: (page 7, 3.4. Line 272)

“MAO activity in 5-ALA-treated fatigued mice showed no statistical significance com-pared to saline-treated fatigued mice (p = 0.2997) (Fig. 5).”

Comments 6

Dose is too high for mice 300 mg/kg/day. Typical Human dose is 0.5-1.7 mg/kg. Test lower doses for translational value.

Response 6

Thank you for pointing that out. “The dosage for mice in this study was determined by converting the human equivalent dose (HED), as outlined in our earlier publications [14]. In pharmaceuticals, the HED is commonly used to calculate the appropriate dose for test animals, considering their body surface area, to achieve effects similar to those in humans. Consequently, the dose of 300 mg/kg/day administered to mice equates to a human equivalent dose (HED) of 24 mg/kg/day, which is within the safety parameters specified in the 5-ALA interview form, which provided by the pharmaceutical company in Japan.” We added these comments in the Materials and Methods section (page 2, line 93).

Reviewer 3 Report

Comments and Suggestions for Authors
  • All the references are old references, and there are new versions that should be used.
  • The used model is strong; hence, can 5-ALA reverse the existing fatigue? I think it is good to test the compound as a treatment in addition to this preventive protocol.
  • The authors should discuss why 5-ALA increased the fat weight.
  • Use colored figures rather than black and white figures.
  • Remove 6. Patents line 418.
  • There are some typo errors.
  • Prefrontal cortex histopathology should be conducted in addition to the protein expression of key proteins.

Author Response

Comments 1

All the references are old references, and there are new versions that should be used.

Response 1

Thank you for pointing that out. We have added a new references as follows.

The dosage for mice in this study was determined by converting the human equivalent dose (HED), as outlined in our earlier publications [14]. (Page 2, line 93)

There are reports indicating that 5-ALA increases SREBP-1c expression, which in turn promotes the biosynthesis of fatty acids and triglycerides [23]. (Page 9, line 340)

On the other hand, there are also reports suggesting that 5-ALA intake reduces fat [24]. (Page 9, line 342)

Comments 2

The used model is strong; hence, can 5-ALA reverse the existing fatigue? I think it is good to test the compound as a treatment in addition to this preventive protocol.

Response 2

Thank you for your comment. As the reviewer 3 pointed out, we also believe that conducting therapeutic studies in animal experiments has value. We will consider this in our future work. “Previous clinical trials demonstrated that chronic treatment with 5-ALA produced, the anti-fatigue effects in healthy subjects who feel chronic physical tiredness suggesting 5-ALA has a potential as a therapeutic agent [2]. On the other hand, this study was conducted using a preventive protocol. We believe that future studies employing a therapeutic protocol are also warranted.” We added these comments in the Discussion section (Page9 4.1 line 315).

Comments 3

The authors should discuss why 5-ALA increased the fat weight.

Response 3

Thank you for your comment. “There are reports indicating that 5-ALA increases SREBP-1c expression, which in turn promotes the biosynthesis of fatty acids and triglycerides [23]. On the other hand, there are also reports suggesting that 5-ALA intake reduces fat [24]. Therefore, it remains unclear whether 5-ALA exerts a fat-increasing or fat-reducing effect. Regarding the present model, further investigation is warranted, including measurements of SREBP-1c, fatty acids, and triglycerides, to clarify these effects.” We added these comments in the Discussion section. (page9, 4.2 Line340)

Comments 4

Use colored figures rather than black and white figures.

Response 4

Thank you for pointing that out. I have changed the figures to color.

Comments 5

Remove 6. Patents line 418.

Response 5

Thank you for pointing that out. We have removed 6. Patents line 418.

Comments 6

Prefrontal cortex histopathology should be conducted in addition to the protein expression of key proteins

Response 6

We appreciate your comment. As pointed by reviewer, “In the present study, histopathological evaluation of the prefrontal cortex, such as assessment of neuronal degeneration or loss, was not performed. Therefore, future studies addressing these aspects will be necessary.” We added these comments in the Discussion section (Page11 line 412). Thank you very much for the helpful suggestion.

Round 2

Reviewer 1 Report

Comments and Suggestions for Authors

Dear authors. The necessary changes have been made. The article is recommended for publication

Reviewer 3 Report

Comments and Suggestions for Authors

The authors followed the comments.